# The Association between Party Horn Use and Respiratory Function in Patients with Dementia: An Experimental Study

**DOI:** 10.3390/medicina59010134

**Published:** 2023-01-10

**Authors:** Misako Higashijima, Hisako Hayashi, Tomotaka Ueda, Yuko Hirano, Hiroyasu Shiozu, Moemi Matsuo

**Affiliations:** 1Faculty of Rehabilitation Sciences, Nishi Kyushu University, 4490-9 Ozaki, Kanzaki 842-8585, Japan; 2Kankikai Healthcare Corporation Tsuji Surgeon Rehabilitation Hospital, 3-24 Ikutamamaemachi, Tennoji Ward, Osaka 543-0072, Japan; 3Unit of Medical Science, Nagasaki University Graduate School of Biomedical Sciences, 1-7-1 Sakamoto, Nagasaki 852-8520, Japan; 4Department of Occupational Therapy, Chubu University, 1200 Matsumotocho, Kasugai 487-8501, Japan

**Keywords:** dementia, party horn, respiratory function, dietary modification, deglutition, Alzheimer’s disease, vascular dementia, rehabilitation, spirometer, aspiration pneumonia

## Abstract

*Background and Objectives:* Respiratory diseases account for 55.5% and 33.1% of all mortality rates in patients with Alzheimer’s disease and vascular dementia, respectively. However, the widespread use of spirometers is often difficult due to challenges in performing the procedure. Therefore, the use of spirometers is usually unfeasible in patients with dementia and hinders the provision of preventive measures for aspiration pneumonia. The party horn is a common toy in many countries and can potentially be used as a novel tool. This study was conducted to analyze the usefulness of the party horn as an assessment tool for respiratory function, and to detect eating-related behavioral problems in patients with dementia. *Materials and Methods*: A total of 62 inpatient participants with dementia (34 males, 28 females; age, mean ± SD, 80.4 ± 7.59 years) were included in the study. The respiratory functions of patients were assessed using a party horn and a spirometer. Assessment items pertaining to cognitive function, mental and behavioral disorders, eating-related behavioral problems, and the required dietary modifications were evaluated to compare between patient groups stratified by respiratory function. *Results*: Significant differences between groups were noted in length of hospital stay, cognitive functions, mental and behavioral disorders, eating-related behavioral problems, and dietary modifications. Forced expiratory volume in 1 s, peak expiratory flow, and eating-related behavioral problems were significantly associated with the party-horn-integrated value (*p* < 0.05). *Conclusions:* Party-horn-based evaluation can facilitate the screening and evaluation of older dementia patients for eating-related behavioral problems and aspiration risk.

## 1. Introduction

Population aging continues to increase rapidly worldwide and is accompanied by a concomitant increase in associated conditions, such as dementia, which has been particularly evident in Japan. Respiratory diseases account for 55.5% and 33.1% of all mortality rates in patients with Alzheimer’s disease and vascular dementia, respectively [1]. Nursing home residents with dementia have a 50% risk of experiencing at least one episode of respiratory disease, such as pneumonia or bronchitis, and an 85.8% chance of eating-related behavioral problems [2]. A previous study on 412 inpatients with dementia in hospital wards reported that 73.2% experienced at least one episode of eating-related problems across the four stages (anticipatory, preparatory, oral, and pharyngeal) of deglutition [3].

Although the elastic recoil of the lungs tends to decrease with age, both deglutition and cough reflexes remain unimpaired in otherwise healthy older individuals [4]. In contrast, older individuals with neurological disorders are highly likely to have impaired swallowing due to which they can develop aspiration pneumonia [4].

The prevention of aspiration pneumonia requires effective expectoration of foreign substances and sputum, which largely depends on the ability to maintain a sufficient expiratory flow rate and forced vital capacity (FVC). However, the widespread use of spirometers for measuring these respiratory functions is unavailable because reliable data acquisition with a spirometer requires sufficient training and experience. Furthermore, it is often difficult to perform this procedure, even in healthy older individuals [5], as it is essential that the examinees comprehend the procedure and fully comply with relevant instructions. Therefore, the use of spirometers is usually unfeasible in patients with dementia and hinders the provision of preventive measures for aspiration pneumonia. To ensure safe eating in patients with dementia, a simple and easy-to-use alternative to the spirometer is required for assessing respiratory function.

The party horn is a common toy in many countries, including Japan, and can potentially be used as a novel tool [6]. A previous study showed a significant difference in the ability to blow the party horn to its full length between patients who used thickening liquids/texture-modifying foods, and those who did not [7]. Another study found a significant correlation between FVC and the duration of party horn extension in patients with dementia [8]. Nonetheless, it is difficult to objectively assess the sustained safe-eating activity of patients with dementia, and no previous study has evaluated the use of alternative assessment tools for respiratory function in this population.

This study was conducted with the aim to analyze the usefulness of the party horn as an assessment tool for respiratory function and detecting eating-related behavioral problems in patients with dementia.

## 2. Materials and Methods

The patients or their family members provided written informed consent, and all methods were performed per the relevant guidelines and regulations.

Study design: the protocol of this experimental prevalence study was approved by the Institutional Ethics Committee of Nagasaki University (approval no. 10072294) and complied with the Declaration of Helsinki (World Medical Association).

### 2.1. Participants

Among the 68 inpatients in the dementia-specialized ward of a psychiatric hospital in Japan who were screened for inclusion in this study, 6 who did not understand how to use the tool or blow the party horn more than 10 cm were excluded. Patients that agreed to participate in this study and could blow the party horn more than 10 cm in the pre-experimental setting were included. Thus, 62 inpatients for whom all required data were available were enrolled in this study (Table 1).

### 2.2. Evaluation Tool

We used a party horn with a maximum length of 90 cm (Figure 1a: Party Horn Entertainment Village, Hyogo, Japan) and a HI-801 spirometer (Figure 1b: Chest M.I., Tokyo, Japan). Before the experiment, an examiner completely unrolled the new party horns three times to minimize unequal resistance inside the 90 cm duct line. The end of the paper tube was then marked with a sticker to easily determine whether the tube had been completely unrolled. Each party horn had a disposable mouthpiece (Tsutsumi Co., Tokyo, Japan). To ensure that the mouthpiece of the party horn had the same diameter as the spirometer, we used customized bullhorn-shaped, black rubber adaptors (Kanae Prosthesis Manufacture, Nagasaki, Japan). The larger end of the adaptor was attached to the spirometer, and the smaller end was connected to the disposable mouthpiece.

### 2.3. Experimental Protocol

The study was conducted as an experimental design. For each patient, a standard examination procedure was followed. Patients were assessed in a private room on days when their condition was stable. First, patients were instructed to blow into their own party horn when seated on a chair or in a wheelchair; this assessment was performed twice with a 3 min rest interval between each respiratory test. Patients were then asked to blow the party horn swiftly and for as long as possible. The maximum length of the party horn was measured by using a 1 cm inter-spatial scale on a table. The time duration from complete unrolling to the point at which the party horn started to resume its coiled form was measured using an HS-80TW stopwatch (Casio, Tokyo, Japan). The “maximum length × party horn extension duration” (the party-horn-integrated value in cm s^−1^) was calculated, and the larger value was used in the analysis.

Next, using a spirometer on another day, the following parameters of respiratory function were examined twice via the vital capacity (VC) test at the point of maximum FVC, which was caused by the opening of the nasal cavity: FVC, percent VC (%VC), forced expiratory volume in 1 s (FEV^1^), FEV^1^ as a percentage of FVC (FEV_1%_), peak expiratory flow (PEF), and forced expiratory time (FET). These data were measured with patients sitting in a chair, using a nose clip to prevent air from coming out of their noses. Data on demographics (e.g., age and sex) and clinical characteristics (e.g., length of hospital stay) were obtained from the medical records. Moreover, cognitive functions, mental and behavioral disorders, eating-related behavioral problems, and dietary modifications were assessed.

Assessment items pertaining to cognitive function (9 items) and mental and behavioral disorders (14 items) were evaluated (Appendix A) in accordance with the instructions provided in the textbook of the nursing certification investigator [9]. Each item was scored according to the following responses: “can” or “no” (2 points); “sometimes” (1 point); and “cannot” or “yes” (0 points). In addition, the assessment form used for the documentation of eating-related behavioral problems (18 items) and required dietary modifications (4 items) is shown in Appendix A. The assessment form for eating-related behavioral problems was evaluated according to two responses: “Yes” = 0 points and “No” = 2 points. The assessment form for the required dietary modifications was evaluated using five responses: “not applicable” = 4 points, “1 item concerned” = 3 points, “2 items concerned” = 2 points, “3 items concerned” = 1 point, and “all items concerned” = 0 points.

### 2.4. Statistical Analysis

To verify the factors that affected respiratory function among older patients, the participants were divided into two groups depending on whether they could blow the party horn to its full length; participants with an integrated value > 2000 cm s^−1^ or FVC > 2 L were assigned to the high group, and the remaining patients were allocated to the low group. The Mann–Whitney U test was used to compare the following factors between the two groups: age, the length of hospital stay, cognitive functions, mental and behavioral disorders, party-horn-integrated value, FVC, %VC, FEV^1^, FEV_1%_, PEF, FET, eating-related behavioral problems, and dietary modifications. Furthermore, multiple linear regression analysis with backward stepwise selection was performed to model the effects of age, cognitive functions, mental and behavioral disorders, eating-related behavioral problems, dietary modifications, FVC, %VC, FEV^1^, FEV_1%_, and PEF (independent variables) on the party-horn-integrated value as the dependent outcome. The corresponding *p*-values, standard regression coefficients (SRCs), and T-values were also calculated. Data were analyzed in SPSS Version 24.0 (IBM, Tokyo, Japan). A *p*-value < 0.05 was set to indicate statistical significance.

## 3. Results

Of the 68 inpatients (36 men, 32 women; age, mean ± SD, 80.8 ± 7.46 years) who were screened for inclusion, 62 (34 men, 28 women; age, mean ± SD, 80.4 ± 7.59 years) were enrolled in this study. Among them, 27 and 35 patients were included in the high and low groups, respectively. There were significant intergroup differences in the length of hospital stay, cognitive functions, mental and behavioral disorders, eating-related behavioral problems, dietary modifications, party-horn-integrated value, FVC, %VC, FEV^1^, and PEF (Table 1).

Multiple linear regression analysis identified three factors that were significantly associated with the party-horn-integrated value: FEV^1^ (SRC −0.325, T 2.949, *p* = 0.005), PEF (SRC 0.283, T 3.912, *p* = 0.005), and eating-related behavioral problems (SRC 0.251, T 2.255, *p* = 0.028) (Table 2).

## 4. Discussion

We found that participants in the low group had a longer hospital stay and lowered cognitive function as compared to those in the high group, which concurs with the findings of previous studies [10]. Similarly, eating-related behavioral problems differed significantly between the groups, implying that eating-related behavior is associated with respiratory function. Restrictive ventilatory impairments were associated with a high risk of aspiration due to the lack of a deglutition apneic period, which is replaced by shallow respiration and tachypnea [11]. Disuse muscle atrophy affects respiration muscles and intercostal muscles, which may lead to restrictive ventilatory impairments [10]. Thus, it is necessary to maintain lung function to ensure safe eating.

However, patients with dementia are often administered antipsychotics to treat disquiet or aggressive symptoms, especially in the early phase of hospitalization. These medications are associated with various side effects, including disuse muscle atrophy secondary to low activity [12]. Previous research suggested that party horns are useful for improving intra-oral pressure [13]. This study showed that party horns could be used as an evaluation tool for the prediction of FEV^1^ and PEF. Our results indicate that FEV^1^ and the party-horn-integrated value have an inversely proportional relationship, whereas PEF and the party-horn-integrated value have a linear proportional relationship. Therefore, the abovementioned findings indicate that the party horn may enable the evaluation of maximum exhalation flow with the longest possible duration. Moreover, we found robust correlations between the integration expiratory pressure and FEV^1^, which have been previously reported [6].

The PEF is proportional to the intention to cough; therefore, party horns can be used as a tool to evaluate eating-associated problems, including aspiration and diminished force of expectoration [14]. The advantage of using party horns is that the evaluation of respiratory function in patients with dementia can be performed more easily, which addresses the challenges caused by limited comprehension ability in patients with dementia to enable the provision of instructions pertaining to deglutition rehabilitation and nutritional guidance. This is pertinent, as pneumonia is a major cause of mortality in patients with advanced dementia [15].

As such, some studies have reported correlations between deglutition and aspiration. The evaluation of respiratory functions with party horns in early-stage dementia may facilitate treatment planning and the provision of individual- or group-level rehabilitative interventions. Such rehabilitation plays an important role in maintaining deglutition and aspiration functions in hospitalized elderly patients [16]. Furthermore, such an assessment can be extended to healthy older adults to ensure safe eating [17].

The limitations of this study include the low reproducibility of data and the lack of a trusting relationship with the examiner due to cognitive impairments of the patients. Thus, whether the results can be generalized to dementia patients with any degree of cognitive impairment remains unclear. Most of these above-discussed issues may be resolved if party horns are incorporated into daily training practices. Additional studies with a big sample size are required to estimate the sensitivity, specificity, and validity of the party horn evaluation system. Furthermore, a power analysis should be used for the determination of the sample size. To investigate the efficacy of respiratory training for patients with dementia, the correlation between time-course changes in respiratory function and eating-related behavioral problems are also should be examined.

## 5. Conclusions

The results of this study support the usefulness of party horns for the evaluation of respiratory functions, such as FEV^1^ and PEF, as well as eating-related behavioral problems. This novel assessment tool may be used to screen and evaluate aspiration risk and eating-related behavioral problems of older patients in the early stages of dementia.

## Figures and Tables

**Figure 1 medicina-59-00134-f001:**
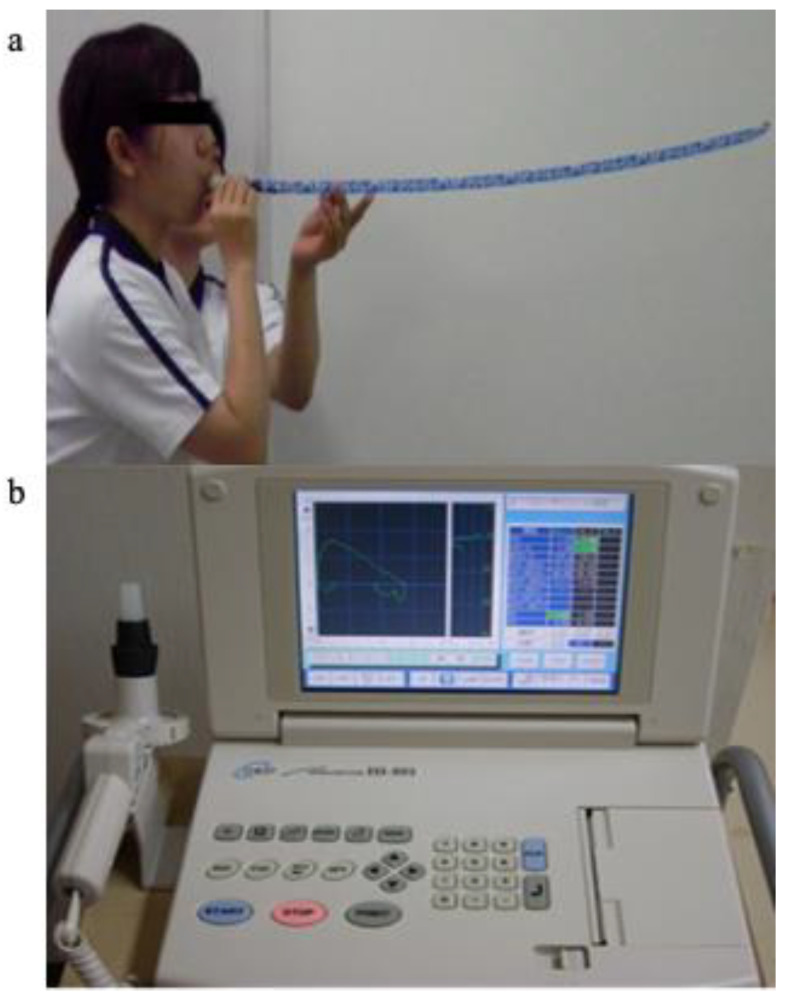
(**a**) Photograph of an individual blowing a 90 cm party horn, and (**b**) HI-801 spirometer (Chest M.I., Tokyo, Japan).

**Table 1 medicina-59-00134-t001:** Comparison of various factors associated with respiratory function between the two groups.

Items	Respiratory Function with a Party Horn		*p*-Value
	High Group (n = 27)	Low Group (n = 35)	
Age (years)	78.7 ± 7.6	81.6 ± 7.1	0.15
Length of hospital stay (days)	117.6 ± 143.5	235.5 ± 226.0	0.04 *
Cognitive functions (points)	14.9 ± 3.0	8.5 ± 4.2	0.001 **
Mental and behavioral disorders (points)	24.1 ± 3.1	21.7 ± 3.9	0.01 *
FVC (L)	2.39 ± 0.5	1.23 ± 0.5	0.001 **
%VC (%)	89.7 ± 18.9	51.9 ± 20.9	0.002 **
FEV^1^ (L)	1.89 ± 0.4	0.98 ± 0.4	0.001 **
FEV_1%_ (%)	78.5 ± 11.4	81.0 ± 16.5	0.42
PEF (L/s)	3.03 ± 1.5	1.66 ± 0.7	0.001 **
FET (s)	5.5 ± 2.6	6.0 ± 4.4	0.91
Eating-related behavioral problems (points)	35.0 ± 3.8	30.2 ± 6.9	0.006 **
Dietary modifications (points)	2.7 ± 1.3	1.4 ± 1.3	0.001 **

* *p* < 0.05 ** *p* < 0.01 by the Mann–Whitney *U* test. Abbreviations: FVC: forced vital capacity; VC: vital capacity; FEV^1^: forced expiratory volume in 1 s; FEV_1%_: FEV^1^ as a percentage of FVC; PEF: peak expiratory flow; FET: forced expiratory time.

**Table 2 medicina-59-00134-t002:** Factors associated with the party-horn-integrated value.

	SRC	T-Value	*p*-Value
FEV^1^	−0.325	2.949	0.005 **
PEF	0.283	3.912	0.005 **
Less severe eating-related behavioral problems	0.251	2.255	0.028 *

* *p* < 0.05, ** *p* < 0.01, by the multiple linear regression analysis with backward stepwise selection. Excluded items = age, length of hospital stay, cognitive functions, mental and behavioral disorders, %VC, FEV_1%_, FET, and dietary modifications. Abbreviations: FEV^1^: forced expiratory volume in 1 s; PEF: peak expiratory flow; SRC: standard regression coefficient; %VC: percent vital capacity; FEV_1%_: FEV^1^ as a percentage of FVC; FET: forced expiratory time.

## Data Availability

The datasets generated and analyzed during the current study are not publicly available due to a lack of participants’ agreement to put the data in public, but are available from the corresponding author on reasonable request.

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
