# Peer review of "The Association between Party Horn Use and Respiratory Function in Patients with Dementia: An Experimental Study"

_medicina, 2023, doi:10.3390/medicina59010134_

Round 1

Reviewer 1 Report

The manuscript by Higashijima and collaborators provides a cost-effective method to assess respiratory function in patients with dementia. The use of party horn toys appears to provide insight into respiratory capacity and could replace the use of spirometry in this cohort of patients where the spirometer is not always a safe option.
Below are my comments and suggestions: 
1) My understanding is that this novel approach was tested previously and that this study builds on the previous findings; however, would it be possible to include an age-matched control group to validate the findings? 
2) Considering the usefulness of this tool, would it be possible to increase the sample size to further validate the findings? Alternatively, could you provide a power analysis for the determination of the sample size?
3) Could you clarify if the patients were sitting or standing up when measuring FVC, %VC, FEV1, FEV1%, PEF and FET?

Minor comment: Please format the manuscript to follow the journal template provided.

Author Response

Response to the comments from the reviewers

We greatly appreciate the comments and suggestions provided by the reviewer, which were very valuable in helping us to further improve the quality of our manuscript. We have revised the manuscript as much as possible, in line with the reviewer’s suggestions. The revised parts are in red colored text. We hope that these corrections and revisions are satisfactory and that the revised version of our manuscript is now acceptable for publication in Medicina.

Response to the comments from reviewer 1

1) My understanding is that this novel approach was tested previously and that this study builds on the previous findings; however, would it be possible to include an age-matched control group to validate the findings?

Response: Thank you for your kind suggestions. We have suggested a potential of usefulness of party horns in our previous research (reference no.6). However, the association with respiratory functions have not been suggested yet. We have just suggested about it in the present research, and we will try to validate it in our future research.  Thus, we have mentioned about this in the limitation section (Page 7, Lines 224-225).

2) Considering the usefulness of this tool, would it be possible to increase the sample size to further validate the findings? Alternatively, could you provide a power analysis for the determination of the sample size?

Response: Thank you for pointing this out. Unfortunately, we cannot increase the sample size for the present research, because not everybody from the hospital agreed to join the study. And we did not use power analysis for the determination of the sample size because we consulted by the previous researches. Hence, we have referred about this in the limitation section (Page 7, Lines 223-226).

3) Could you clarify if the patients were sitting or standing up when measuring FVC, %VC, FEV1, FEV1%, PEF and FET?

Response: We have added the relevant information in the Experimental protocol section (Page 4, Line 127).

Reviewer 2 Report

RE: medicina-2092334:

Respiratory diseases account for about half of all mortality rates in patients with Alzheimer’s disease and vascular dementia, respectively. However, the widespread use of spirometers is often difficult due to the performing procedure. Therefore, the use of spirometers is usually unfeasible in patients with dementia and thereby hinders the provision of preventive measures for aspiration pneumonia. The authors assessed the usefulness of the party horn as an alternate assessment tool for respiratory function, and to detect eating-related behavioral problems in patients with dementia.

Sixty-two inpatient participants with dementia were included in the study. The respiratory functions of patients were assessed using the party horn and spirometry. Assessment items pertaining to cognitive function, mental and behavioral disorders, eating-related behavioral problems, and the required dietary modifications were evaluated to compare between patient groups stratified by respiratory function. Results: Significant differences between groups were noted in the length of hospital stay, cognitive functions, mental and behavioral disorders, eating-related behavioral problems, and dietary modifications. FEV1, FVC, peak expiratory flow, and eating-related behavioral problems were significantly associated with the party horn integrated value (p<0.05).

Conclusions: Party horn-based evaluation can facilitate the screening and evaluation of older dementia patients for eating-related behavioral problems and aspiration risk.

Comments:

1. The paper is well-written with appropriately constructed background, hypothesis, methods, results, discussion and conclusion sections.  The references are appropriate and relevant to the research. The use of the party horn is a clever, inexpensive and simple means of assessing respiratory function in neurologically impaired patients and a potential alternative to spirometry.

2.  P. 2, ll. 59-60:  As an alternative, have the authors considered use of impulse oscillometry (IO)? It is a sensitive test to detect changes in respiratory compliance and resistance, easy to use, noninvasive and requires no patient effort. The use of IO coupled with pulse oximetry can easily pick up structural lung changes related to aspiration.

3. P. 4, ll. 117-124: Was a nose clip applied during spirometry?

4. P. 4, Statistical Analysis:  In addition to analysis described here, one or 2 plots showing the correlation between horn duration and FVC, FEV1, PEF and FEV1/FVC would provide nice graphical representation shown in Table 2. The r- and p-values can be added to the graphs.

Author Response

Response to the comments from the reviewers

We greatly appreciate the comments and suggestions provided by the reviewer, which were very valuable in helping us to further improve the quality of our manuscript. We have revised the manuscript as much as possible, in line with the reviewer’s suggestions. The revised parts are in red colored text. We hope that these corrections and revisions are satisfactory and that the revised version of our manuscript is now acceptable for publication in Medicina.

Response to the comments from reviewer 2

  1. The paper is well-written with appropriately constructed background, hypothesis, methods, results, discussion and conclusion sections. The references are appropriate and relevant to the research. The use of the party horn is a clever, inexpensive and simple means of assessing respiratory function in neurologically impaired patients and a potential alternative to spirometry.

Response: We appreciate about your comments. We will try to show more usefulness of party horns in our future research as well.

  1. P. 2, ll. 59-60: As an alternative, have the authors considered use of impulse oscillometry (IO)? It is a sensitive test to detect changes in respiratory compliance and resistance, easy to use, noninvasive and requires no patient effort. The use of IO coupled with pulse oximetry can easily pick up structural lung changes related to aspiration.

Response: Thank you for your kind suggestions. Unfortunately, we don’t have an IO, and didn’t considered use of it. Because we wanted to use the party horn which is very traditional toy for Japanese. However, we will learn about IO and try to educate ourself more.

  1. P. 4, ll. 117-124: Was a nose clip applied during spirometry?

Response: We have added more information in the Experimental protocol section (Page 4, Lines 126-127).

  1. P. 4, Statistical Analysis: In addition to analysis described here, one or 2 plots showing the correlation between horn duration and FVC, FEV1, PEF and FEV1/FVC would provide nice graphical representation shown in Table 2. The r- and p-values can be added to the graphs.

Response: In this research, we used Multiple linear regression analysis which automatically pick the factor which significantly correlate with the party horn integrated value. Thus, FVC, FEV1 and FEV1/FVC were technically have cut off from the analysis result. The SRC in the table indicate the strength of the correlation in this analysis; this is representing the r-value. We have added p-value in the table (Page 5, Line 172).
